# Experiences of Indigenous peoples living with pelvic health conditions: A scoping review

Kaeleigh Brown[1]*, Katherine Choi[2], Esther Kim[3], Sandra M. Campbell[4], Jane Schulz[5], Pertice Moffitt[6], Susan Chatwood[1,3]

1 University of Alberta School of Public Health, Edmonton, Alberta, Canada, 2 Government of Nunavut, Iqaluit, Nunavut, Canada, 3 Institute for Circumpolar Health Research, Yellowknife, Northwest Territories, Canada, 4 University of Alberta Libraries, Edmonton, Alberta, Canada, 5 University of Alberta Faculty of Medicine & Dentistry, Edmonton, Alberta, Canada, 6 Aurora Research Institute, Yellowknife, Northwest Territories, Canada

* kaeleigh@ualberta.ca

## Abstract

### Background

Pelvic health conditions significantly impact quality of life and are prevalent in the general population. Urinary and fecal incontinence, pelvic organ prolapse, and pelvic pain are examples of pelvic health conditions. A scoping review was conducted to understand what is currently known about pelvic health conditions experienced by Indigenous populations worldwide. To date, no such review has been reported.

### Methods

A scoping review methodology was used. In February 2024, a search was conducted, capturing both primary and grey literature. An iterative process of abstract and full text screening was conducted by two reviewers before proceeding to data extraction. Inclusion criteria focused on English publications and reports of pelvic health conditions experienced by Indigenous peoples. Data was collected in Google Sheets, and then underwent descriptive statistical analysis. Publications that provided qualitative data were further analyzed using thematic analysis.

### Results

A total of 242 publications were included in the analysis. Several patterns emerged: most publications originated from English-speaking regions, fewer than half of publications specifically recruited Indigenous peoples, women participated in more studies than men, and bladder conditions were most frequently reported. Perceptions of pelvic health conditions and experiences with help seeking and the health care system were described. Notable gaps were a lack of publications and representation of Indigenous peoples from China, Russia, and Nordic countries, minimal representation of gender diverse populations, few publications reporting on auto-immune and bowel conditions, and limited mention of trauma-informed and culturally safe approaches.

**Data availability statement:** All relevant data are within the paper and its Supporting Information files.

**Funding:** KB received funding support from the Alberta SPOR Support Unit Graduate Studentship (https://absporu.ca/) while working on this study. EK was funded by the CIHR NWT-Network Environments for Indigenous Health Research (https://cihr-irsc.gc.ca/e/51161.html) as a reviewer. JS has received an honorarium for a talk from AbbVie Canada. The funders had no role in study design, data collection and analysis, decision to publish, or preparation of the manuscript. There was no additional external funding received for this study.

**Competing interests:** The authors have declared that no competing interests exist.

## Conclusions

This study highlights gaps in the current literature around gender representation, bowel and auto-immune conditions, regional representation, and the use of safety frameworks, which may inform future research initiatives. It also summarizes the existing literature, which may inform clinical and health system-level decision making.

## Introduction

The United Nations (UN) estimates that there are at least 370 million Indigenous peoples across the globe [1]. Indigenous peoples is a broad term used to describe the original inhabitants of a region prior to the arrival of other peoples [1]. To increase understanding of the term "Indigenous", the UN outlines several criteria, rather than offering an official definition. These criteria include: self-identification, relationships with the land, association with pre-colonial/settler societies, distinct knowledges (e.g., culture, language, social systems), belonging to non-dominant groups of society, and commitment to continue as "distinctive peoples and communities" [1]. Indigenous peoples experience negative health outcomes when compared to non-Indigenous populations, and there have been calls from national [2–4] and international [5,6] fronts to address these attitudes and inequities. Negative health outcomes include higher rates of chronic health conditions [7,8] and mortality [8,9]. Disparities in the social determinants of health (e.g., housing, employment, income) are also prevalent [7]. These inequities have been attributed to several factors including racism and colonization [7,8,10,11]. Indigenous peoples also report barriers to accessing health care services [7,10,11]. In general, individuals access health care not just for chronic or emergent medical needs, but for other concerns, like pelvic health conditions, which impact quality of life.

The pelvic floor includes interconnections between many body systems which are housed within the bony pelvis. These structures are supported by connective tissue, ligaments, and the pelvic floor muscles. When there is a change in these supporting structures, whether due to aging, pregnancy and childbirth, nervous system sensitivity, or pathology, an individual may experience symptoms of a pelvic health condition. Symptoms of pelvic health conditions in men include erectile dysfunction, in women include protrusion of pelvic organs (prolapse), and in both genders include urinary or bowel incontinence/urgency/frequency or pain (with voiding, defecation, or intercourse). Several interventions exist for most pelvic health conditions, including medication, pelvic floor rehabilitation, surgery, continence products, pessaries, and behavioural modifications.

Pelvic health conditions are prevalent in the general population [12–16] and the number of individuals affected is expected to increase [17]. The existing literature demonstrates varying prevalence rates across race/ethnicity [18,19], particularly with an over-representation of White people and an under-representation of Indigenous people and people of colour in pelvic health research [20–22]. This may have impacts on clinical decision making and generalizability of research results [20–22].

To date, there has been no published review of pelvic health conditions experienced by Indigenous populations. Due to this gap, it is unclear whether Indigenous peoples experience similar inequities in pelvic health as they do with other health conditions. The purpose of this scoping review is to describe what is currently known about Indigenous peoples' experiences with pelvic health conditions by asking the question: "What does the literature say about pelvic health conditions experienced by Indigenous populations worldwide?". Because there is a need for exploratory work in this area, we decided a scoping review method was best suited.

This scoping review will identify gaps and patterns in the existing literature and provide recommendations for future research and practice [23].

## Methods

We selected a scoping review methodology to systematically search and analyze the existing literature on pelvic health conditions experienced by Indigenous peoples worldwide. This methodology was a good fit for the topic, since it has yet to be systematically explored, and because the scope of the research question was broadened beyond interventional studies [24]. Although our search strategy was developed in advance, we did not register the scoping review protocol. We acknowledge that this appears to reduce transparency of the review [25]. However, this allowed us to be responsive to the literature, recruit an additional reviewer (EK), and adapt our inclusion and exclusion criteria in an iterative manner. As per the PRIMSA-ScR checklist (S2 Checklist), our search strategy, dataset, and sorting criteria are included in this publication to allow for replication of our results. Several authors have explored and further defined scoping review methodology (see Arksey & O'Malley [24], Levac et al [26], and Bradbury-Jones et al [23]). Our review was informed by all three publications.

### Search strategy

A search was executed by an expert searcher/health librarian (SMC) from the University of Alberta, Edmonton, Canada, on the following databases: PROSPERO, OVID Medline, OVID EMBASE, OVID Global Health, Wiley Cochrane Library (CDSR and Central), EBSCO CINAHL, Proquest Dissertations and Theses Global and SCOPUS using controlled vocabulary (e.g.,: MeSH, Emtree, etc.) and key words representing the concepts "pelvic health conditions", "Indigenous People". Variants of search filters were applied for "Indigenous People" [27–40]. Search terms for pelvic health conditions were developed in 2022, and informed by the International Continence Society's Incontinence, 6th Edition [41], the first author's (KB) clinical experience, and reviewed by a practicing gynecologist (JS). Searches were adjusted for each database. No limits were applied. Databases were searched from inception to mid-February 2024. Detailed search strategies are available in S1 Appendix.

Reference lists of the included full texts were also searched to identify any articles that may meet the inclusion criteria. In one case [20], the corresponding author was contacted to acquire a list of citations used in their analysis. Additionally, hand searching of key journals (BMC Women's Health, Journal of Women's & Pelvic Health Physical Therapy, and International Urogynecology Journal) was conducted from inception to January 31, 2024. Two organization websites (World Health Organization and International Continence Society) and an internet search engine (DuckDuckGo) were also searched to identify additional resources from the media, organizations, and conference presentations.

Once the results were uploaded to Covidence [42], two rounds of review were conducted by two reviewers (either KB and KC, or KB and EK): first screening the abstracts and titles, followed by screening the included full text resources. Screening criteria for each round are outlined in Table 1. Publications were included based on reporting of pelvic health conditions specific to Indigenous participants in the results. There were no criteria based on the quality or rigor of the publication, nor the selected methodology. If there was a discrepancy between the two reviewers, they met to discuss and come to a consensus on whether to exclude or proceed with inclusion. This was an iterative process, where reviewers modified the screening criteria based on the search results and during data collection, where the co-authors identified literature that did not meet the inclusion criteria and were subsequently excluded. For example, our search terms included gynecological, prostate, and colorectal cancers, which, through the course of treatment, may result in the development of

**Table 1. Scoping review inclusion and exclusion criteria.**

| Abstract Review | |
| --- | --- |
| *Inclusion Criteria* | |
| 1. Title and/or abstract written in English | |
| 2. Pelvic health condition and/or urological condition and/or gynecological condition and/or anorectal condition mentioned in title and/or abstract | |
| 3. Indigenous people/population mentioned in title and/or abstract | |
| **Full Text Review** | |
| *Inclusion Criteria* | *Exclusion Criteria* |
| 1. Full text in English | 1. No report of Indigenous population |
| 2. Indigenous population reported AND their experience/prevalence/etc. with pelvic health condition also reported | 3. No report of pelvic health condition |
| | 4. Unable to access or acquire full text |

a pelvic health condition. These publications were usually excluded due to a lack of reporting on pelvic health conditions.

## Data collection

Data extraction, also known as charting the data [24], is an iterative process, which evolves as the reviewers become more familiar with the literature [26]. To ensure extraction was consistent between reviewers, KB, KC and EK extracted data from up to ten articles, and then met to share their findings [26]. Any discrepancies or disagreements were discussed between reviewers. Data extraction continued, and a Google Sheets spreadsheet was used to collate the results. Additional meetings between reviewers occurred ad hoc, to ensure data extraction was responsive to the literature and maintained consistency among the reviewers. The extracted data included: population demographics (e.g., age, parity, medical history), population socioeconomic factors, publication characteristics, region information (e.g., country, targeted recruitment of Indigenous peoples), pelvic floor information (e.g., epidemiology, conditions reported, interventions, help-seeking behaviours, described experiences with pelvic health conditions), and safety information (cultural safety, trauma-informed approach). These headings were developed from an existing template used in previous scoping reviews by the final author (SC), and then modified to reflect the pelvic health and Indigenous health literature.

Indigenous people have experienced and continue to experience discrimination and harm in healthcare and research [43]. Culturally safe and trauma-informed practices have been proposed to improve safety in healthcare. We believed it was important to include headings for cultural safety and trauma-informed practices/frameworks to understand the extent of their implementation in Indigenous pelvic health research. Cultural safety is a social justice approach to healthcare based in Māori Indigenous knowledges [44,45]. It addresses power dynamics between patients and healthcare providers and has been adopted internationally [45]. The concept within healthcare has been interpreted in several ways [45], but its use within research has not been explored extensively [44].

Traumatic experiences are prevalent, such as around birth and sexual trauma [46,47], and there is a statistical association between sexual trauma and some pelvic health conditions [48]. Trauma Theory suggests that traumatic experiences can affect how the body functions, including mental and physical health [49,50]. Researchers have theorized that adopting a trauma-informed approach will have a positive effect on service users and participants [50,51]. Trauma-informed practice has been implemented in clinical and educational practices for a number of years [52]. Providers are encouraged to implement trauma-informed practice as a standard precaution with all clients, and "rely[] on procedures that are most likely to be

growth-promoting and least likely to be retraumatizing" [53]. This may look like providing clients with relational collaboration, a safe environment, validation, and opportunities for choice and control [53]. Several authors have contributed to trauma-informed scholarship within Indigenous contexts by: critiquing existing approaches to educational research [52], developing new frameworks for patient engagement [54], summarizing existing health literature [55], and sharing culturally safe and trauma-informed recommendations for practitioners working with specific populations [56].

### Analysis

The PAGER framework guided analysis and reporting of the literature review findings [23]. The five domains of the framework are: Patterns, Advances, Gaps, Evidence for practice, and Research recommendations. Bradbury-Jones et al [23] recommend using a patterning chart to identify trends in methodology and themes. Due to the number of included publications, we opted out of this process, and instead looked for patterns within the charted data around publication characteristics, participant demographics, regions, pelvic health conditions, Indigenous representation, and safety. Following pattern identification, we then proceeded to examine the evolution of the research – specifically looking at theoretical and methodological advances [23]. To identify gaps in the literature, interconnections between themes and the current research and practice landscape were considered [23]. Evidence for practice and research recommendations were considered at the end of analysis to provide potential knowledge users with information, and guidance on what the next research steps may be [23,26].

## Results

Results (2911) were exported to Covidence review management software [42], where duplicates (453) were removed. Publications were sourced from electronic databases (2799), journal hand searching (81), DuckDuckGo (11), and citation searching (20). Nineteen additional duplicates were removed manually. After abstract and full text screening, 18 publications were merged due to duplication (e.g., same content published in several conference abstracts, or conference abstract merged with original journal article). The inclusion criteria used to screen abstracts was broader than that of full text publications. Aside from a requirement to be published in English, abstracts that mentioned a potential pelvic health condition (e.g., bladder issues) and Indigenous populations were included. Full text publications were required to meet an additional criterion, in that Indigenous people must also be experiencing a pelvic health condition. For example, some excluded publications reported Indigenous participation in a study, but they were part of a control group and were not experiencing a pelvic health condition. This left 242 publications for analysis. See Fig 1 for a summary of the study selection process, as illustrated in a Preferred Report Items for Systematic Review and Meta-Analysis (PRISMA) flow chart [25].

### Publication characteristics

Fig 2 provides a summary of all included studies based on region of publication. North America was represented most frequently (49.5% of publications), whereas the Middle East (1.7%) and multinational (1.2%) publications were the least represented. To provide more regional representation, the Middle East and Indian sub-continent were represented separately from their respective continents, and publications from Mexico were included in Latin America and excluded from North America. Notably, there were two publications from circumpolar nations (Northwest Territories, Canada; Alaska, United States of America), which includes Nordic countries, Russia, Greenland, Alaska, and the northern territories of Canada. The

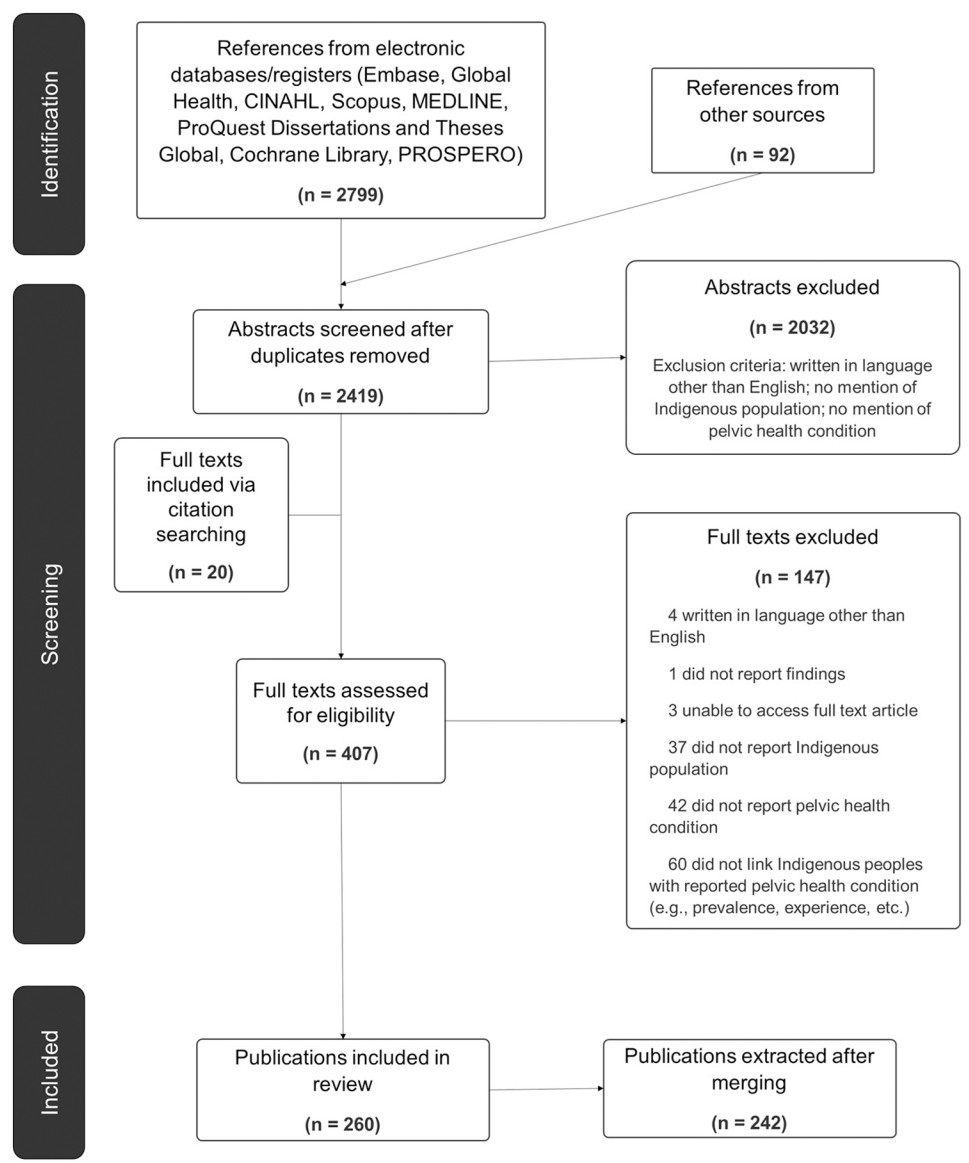

**Fig 1. PRISMA flow chart.**

majority of publications were from the primary literature (80.2%) and utilized quantitative methodologies (68.6%).

The proportion of Indigenous peoples in the publications varied widely, from 0-100%. The proportion of Indigenous participants was reported 303 times (this is due to multiple treatment arms in interventional studies, or several Indigenous groups identified within a publication – where the total did not equal 100%). As shown in Fig 3, the distribution of representation weighed heavily towards 100% (24.4% of instances), and towards < 5% (21.8%).

Approximately 48% of references specifically recruited Indigenous peoples (n = 116). These references were then isolated and underwent subsequent analysis. Per Fig 2, the distribution of publication region was highest in Australia/New Zealand (25.8%), followed by Africa (22.4%), and the Indian sub-continent (18.1%). As with the original sample, the majority of

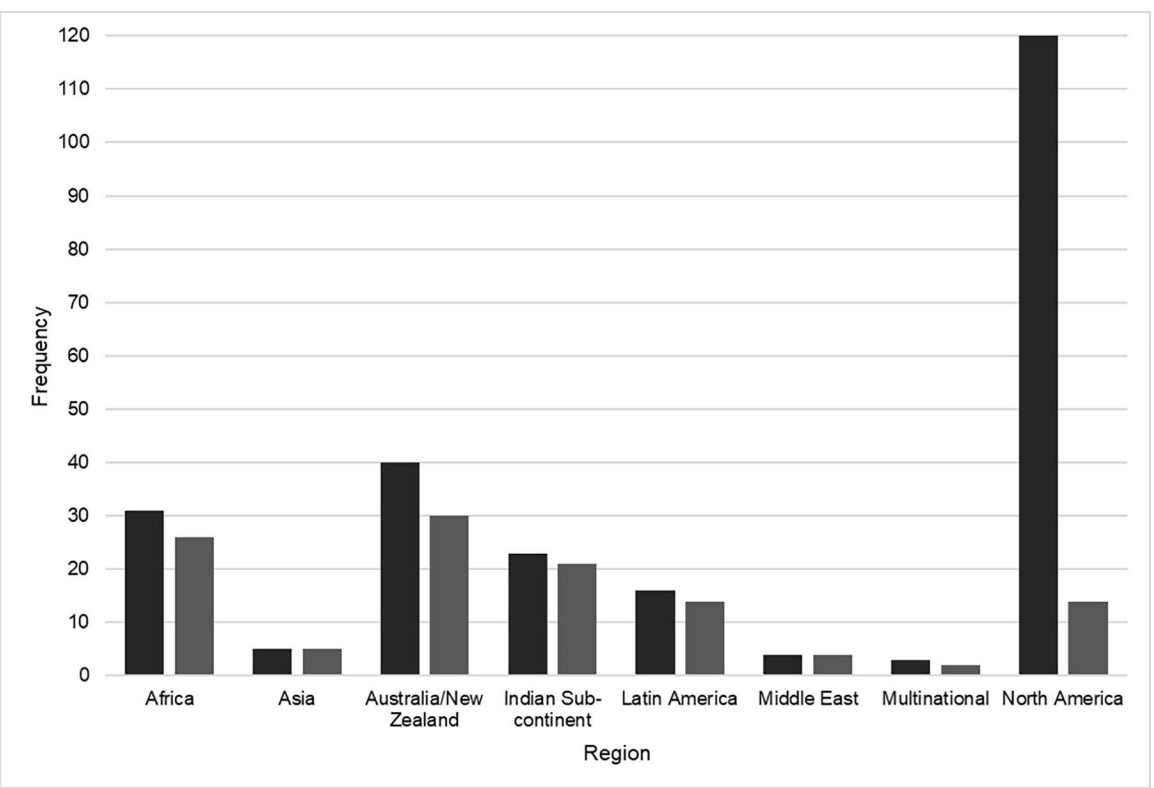

**Fig 2. Regional representation of included publications.** (Dark grey) All included full text. (Light grey) Indigenous recruitment only.

publications were from the primary literature (83.6%). The methodology of the publications was distributed as such: quantitative (49.1%), ethnobotanical/zoological (21.6%), qualitative (9.5%), reviews (systematic, scoping, narrative, etc. – 7.8%), mixed methods (4.3%), other (1.7%). The remaining publications (6.0%) either did not report the methodology, or it was not applicable (e.g., news article). In this sample, the proportion of Indigenous participants was reported 123 times. Publications that specifically recruited Indigenous populations, and attained 100% of Indigenous participants, were in the majority (60.2%). This differed from the initial sample, as illustrated in Fig 3. One publication did not recruit Indigenous participants but spoke with non-Indigenous practitioners who provided services to Indigenous peoples.

Publications were analyzed by year, as visualized in Fig 4. Fewer than five publications were released annually between 1960 and 2008. Starting in 2003, and each year until 2024, at least one publication specifically recruited Indigenous people. It should be noted that in Fig 4, 2024 is to be considered an incomplete year, as we ran our search in February of 2024.

## Demographics

Women/females were represented in 45.6% of publications, whereas two-spirit people were represented in < 1%. Men were included in 10.3% of the publications, 28.4% of the publications included men and women, and 14.7% did not report on sex or gender or it did not apply (i.e., the publication reported on service delivery for Indigenous populations). The reported age of Indigenous participants varied, both in the method of reporting (e.g., median, mean, range) and in the represented ages (children to elders). Analysis was challenging, due to this

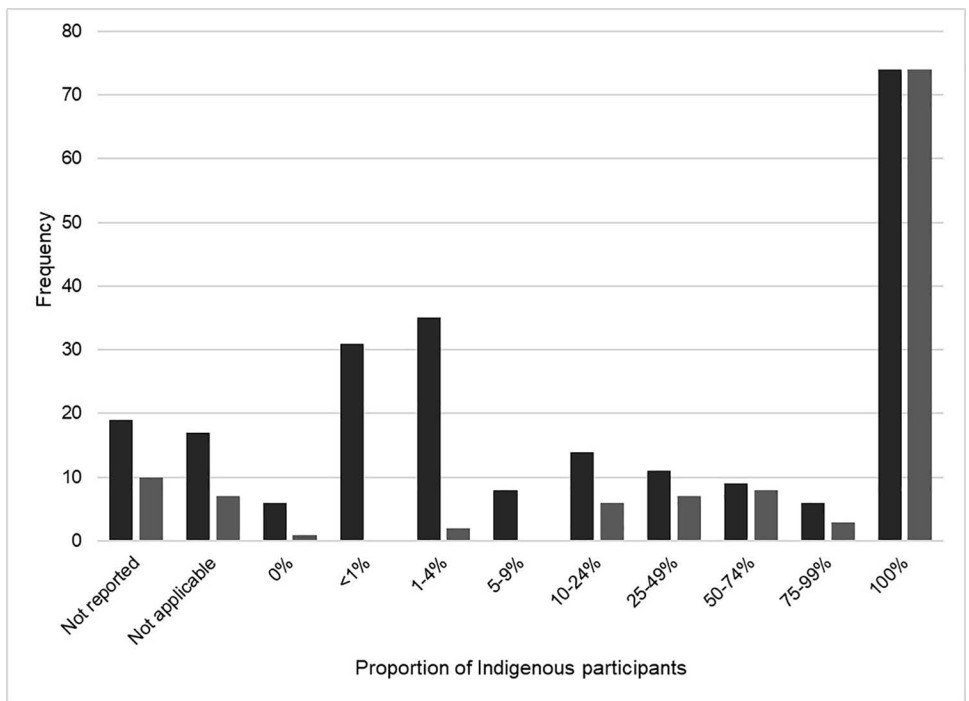

**Fig 3. Frequency of reporting of the proportion of Indigenous participants.** (Dark grey) All included full text. (Light grey) Indigenous recruitment only.

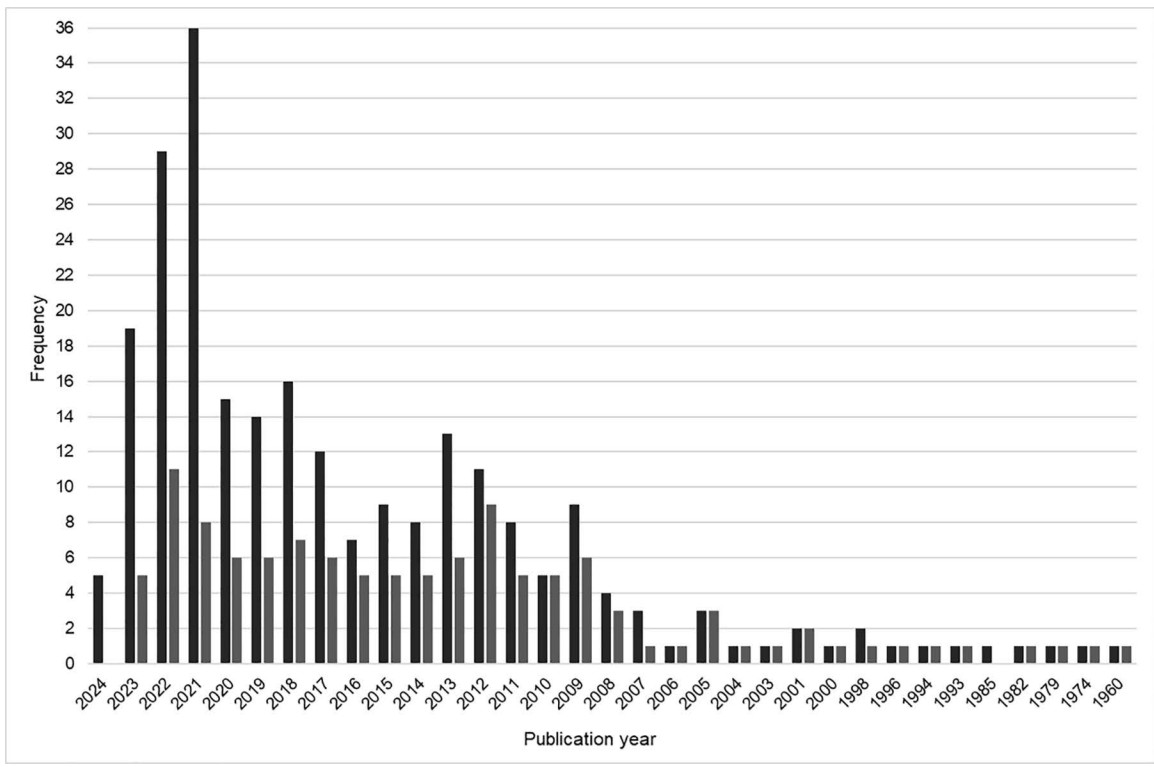

**Fig 4. Frequency of publications by year.** (Dark grey) All included full text. (Light grey) Indigenous recruitment only.

variability in reporting. We opted to aggregate the data into categories (child, youth, adult, Elder) based on our best estimate on what was reported and extracted. Not all publications reported age, leaving 83 publications for analysis. The majority (91.3%) of those publications recruited adults (ages 19-59), which includes articles that reported age ranges extending into youth (ages 12-18) and Elder (ages 60+) categories. Only 6.2% publications recruited Elders, and 2.5% only recruited children (ages 0-11) or youth.

### Safety

The use of trauma-informed approaches/frameworks and cultural safety practices was analyzed. Of the analyzed articles (n = 116), 2.6% (n = 3) reported using a trauma-informed approach or framework and 25.9% (n = 30) mentioned the use of culturally safe practices. Of the three publications that reported using a trauma-informed approach or framework, 100% were from Australia and from the grey literature (one report, one conference presentation, and one clinical guideline). Seventeen (56.7%) publications that mentioned cultural safety were also from Australia or New Zealand. The remaining publications were from Latin America (16.7%), the Indian sub-continent (6.7%), Africa (6.7%), North America (6.7%), or from multinational locations (6.7%). Most of these publications (80%) were from the primary literature.

### Pelvic health information

Of the articles that were interventional or described treatment options (n = 39), the majority (66.7%) described herbal or traditional medicines, 15.3% mentioned surgical treatment, 7.7% education or social support, 5.1% a combination of Western and traditional medicines, and the remainder mentioned medications or multiple treatment options (2.6% each). A regional summary of interventions is represented in Fig 5. All publications from the Indian sub-continent (n = 17) described herbal or traditional medicines. Publications from Africa (n = 11) reported the use of Western-based treatments (e.g., surgery, medications, multiple treatment options) (n = 6), herbal or traditional medicines (n = 4), or a combination of Western and

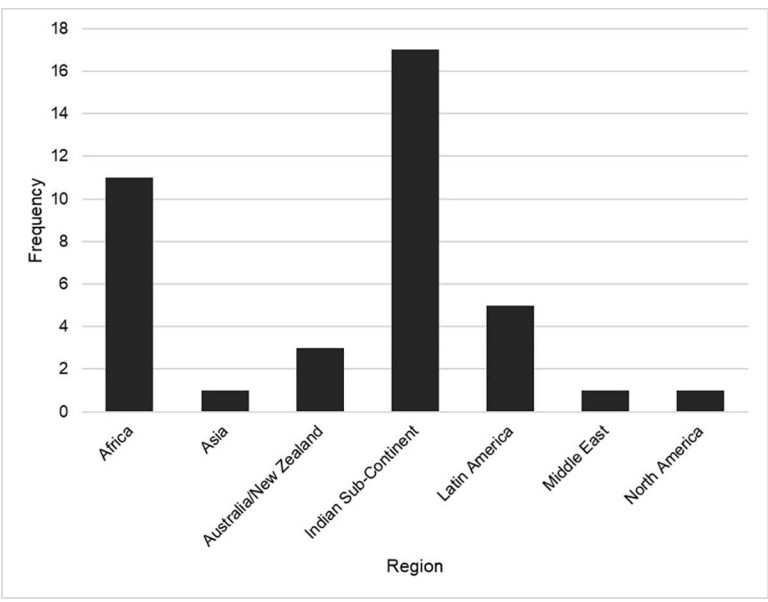

**Fig 5. Distribution of described pelvic health condition treatments by region.**

traditional medicines (n = 1). Of the five publications from Latin America, three described the use of traditional or herbal medicines, one reported a combination of Western and traditional medicines, and another described education or social supports. Australian publications (n = 3) reported social supports or education (n = 2) or Western-based treatment (n = 1). The singular publications from Asia, the Middle East, and North America reported the use of herbal or traditional medicines, Western-based medicine, and herbal or traditional medicine, respectively.

Fig 6 visualizes the frequency of pelvic health conditions mentioned in the literature that focused specifically on Indigenous peoples. One publication did not report a pelvic health condition but spoke to pelvic health physiotherapy service delivery to Australian Indigenous peoples. It was removed from this analysis (n = 115). Since many publications reported multiple pelvic health conditions, the total frequency is higher than the number of publications. Bladder conditions (including stress, urge, and mixed incontinence, nocturia, bladder retention, lower urinary tract symptoms, bladder urgency, bladder frequency, and post-micturition dribble) were reported most frequently (37.2%), followed by pain conditions (dysuria, dyspareunia, dysmenorrhea, endometriosis, and pelvic pain) (24.2%), pelvic organ prolapse (13.0%), and erectile dysfunction (11.1%). Publications that reported incontinence, but did not further define the term (i.e., bladder and/or bowel incontinence) were grouped under the label "bladder or bowel incontinence". Fistulas were categorized separately from bladder and bowel conditions. Although a fistula may result in leakage of urine or feces, the mechanism differs from other incontinence conditions.

We also extracted epidemiological data, including prevalence of pelvic health conditions. This is illustrated in Fig 7. There was wide variability in the prevalence of urinary incontinence (including stress, urge, and mixed incontinence), pelvic organ prolapse, and erectile dysfunction. The large variability can be explained by several factors including severity of

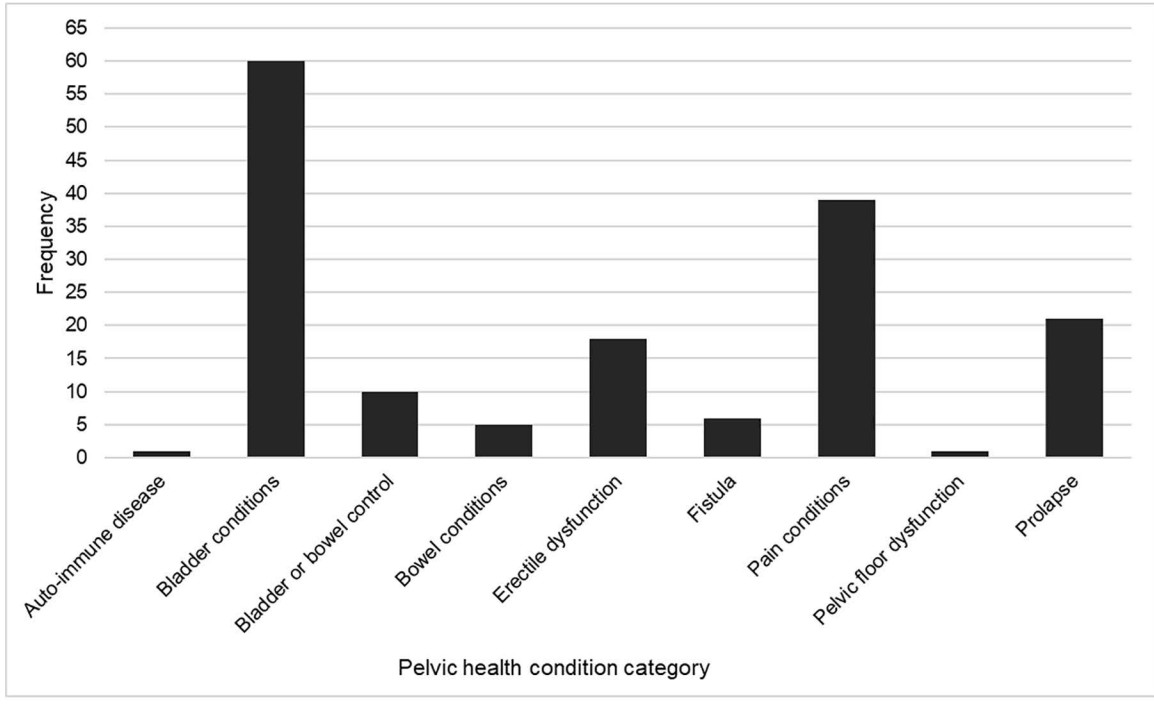

**Fig 6. Frequency of pelvic health conditions described in the literature.**

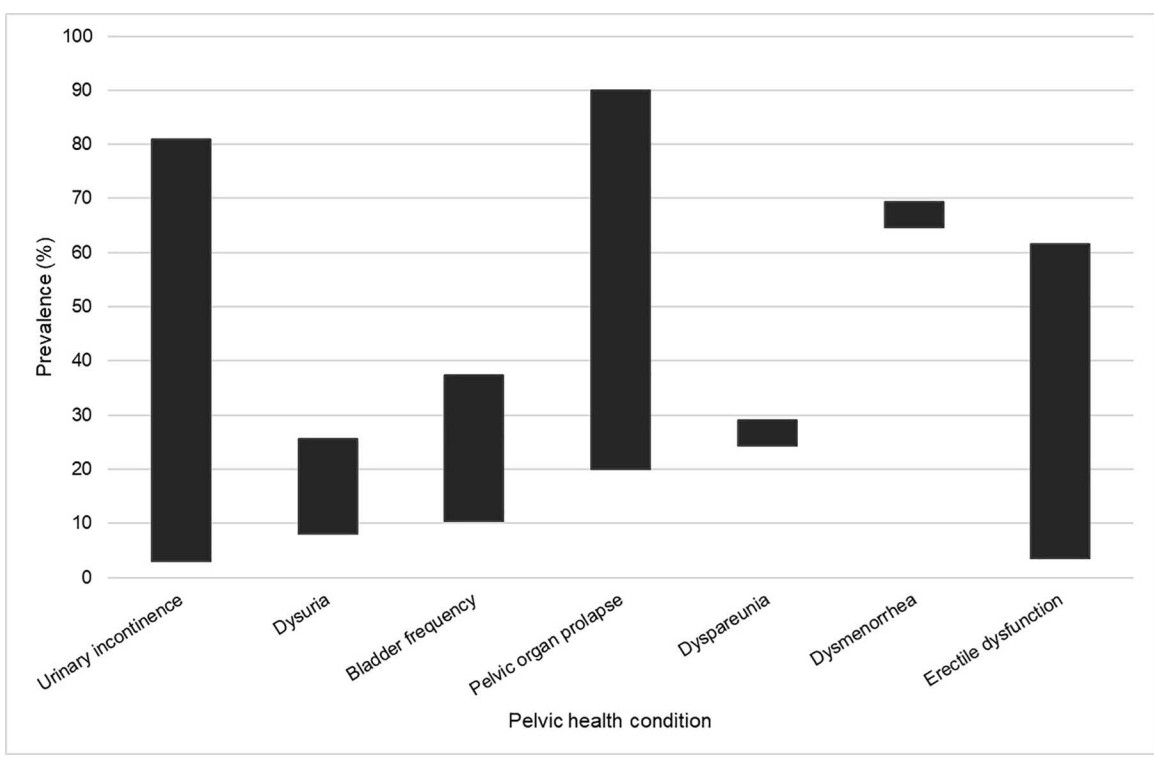

**Fig 7. Prevalence of specific pelvic health conditions reported in the literature.**

symptoms (e.g., inclusion of mild vs severe erectile dysfunction, grade of pelvic organ prolapse), life course (e.g., pregnancy vs post-partum, peri- vs post-menopausal participants), structure causing the symptoms (e.g., uterocele, cystocele, rectocele), and symptom presentation (e.g., urge incontinence, stress incontinence, mixed incontinence). Narrower ranges were reported with dyspareunia and dysmenorrhea. There was one report of accidental bowel leakage (fecal incontinence) experienced by 2% of older Indigenous adults [57].

## Help-seeking and service delivery

Barriers and facilitators to help-seeking and service delivery were identified in the publications. Cultural norms and traditions in some cases were facilitators to accessing treatment, such as remedies rooted in Indigenous knowledges (n = 9), or in one publication, opportunities to seek care outside of community for Indigenous men. In other cases, cultural norms were a barrier to seeking help (n = 5). Privacy, gender roles and expectations, and societal perceptions were reported as limiting factors.

Eighteen publications described the influence of the healthcare system and its staff, in many cases, posing a barrier to access. Service costs, location of services in relation to communities, healthcare staff attitudes and behaviours, timeliness of service, and staff competencies were reported as barriers. Two publications reported efforts to facilitate access to services, either by expanding the scope of practitioners or by addressing bureaucratic barriers experienced by practitioners.

Finally, emotions, knowledges, and beliefs of the participants in relation to help-seeking and service delivery were also described as barriers and facilitators. Shame, shyness, stigma, embarrassment, and self-perception were barriers (n = 7). Four publications described active

knowledge seeking, and participants expressing interest in accessing services or learning more about pelvic health conditions as facilitators.

### Perceptions of pelvic health conditions

Similar themes regarding emotions and knowledge seeking arose around how people perceived pelvic health conditions. Unpleasant emotions such as embarrassment and shame were reported (n = 4). Knowledge seeking and awareness was identified as a need (n = 4) and in some cases, was being addressed by researchers, health care providers, and Indigenous community members (n = 5).

Participants reported several reasons for their symptoms (n = 7), with some rooted in spiritual beliefs (n = 3). Moreover, personal, interpersonal, and societal impacts of these pelvic health conditions were described, such as leading to tension within marriages (n = 2), need for family and community support (n = 5), migration away from communities to larger centres for treatment (n = 1), loss of fertility (n = 2), reduction in quality of life (n = 1), health outcomes (n = 2), and participation in work and community (n = 2).

### Impact of healthcare systems

Functioning of the healthcare system as a whole was another theme that was mentioned across the included publications, in relation to pelvic floor health. Experiences seeking treatment were described, many of which shared stories of system barriers and maltreatment by healthcare providers (n = 5). Gaps in services and training were identified (n = 5). Solutions around provision of accessible and safe treatment by expanding the scope or responsibilities of healthcare providers, were also described (n = 2).

## Discussion

This review systematically searched the literature to understand what is currently known about pelvic health conditions experienced by Indigenous peoples worldwide. The findings of this review will be summarized and discussed within the PAGER framework proposed by Bradbury-Jones et al [23].

### Patterns

Much of pelvic health literature originated from English-speaking regions – namely North America, Australia, and New Zealand. However, when we examined articles that specifically recruited Indigenous peoples, North American publications moved from most frequent to mid-range. To our knowledge, international trends around the country/region of publication origin has not been explored in the pelvic health literature.

Women (i.e., sex reported as female or gender reported as woman) participated in more studies than men (i.e., sex reported as male or gender reported as man) or gender diverse people. To our knowledge, there has been no study examining sex and gender representation within the pelvic health literature. Our finding is not surprising, considering that pelvic health conditions such as incontinence are more prevalent in women [12,58], female specific conditions (e.g., endometriosis, pelvic organ prolapse) were more frequent in our data, and that our journal hand search focused on publications relating to women's health.

Bladder conditions were most frequently reported in the publications, whereas autoimmune diseases, bowel conditions, and fistula were least frequently reported. As in the broader literature, this review also demonstrated variability in prevalence rates of pelvic health conditions. For example, Vogel et al [13] report post-partum urinary incontinence ranging between 8-31%, Shaw et al [12] report 23.7% of Canadian adults, and Peinado-Molina et al [15] report

55.8% experienced by adult women in Spain. Our study reports urinary incontinence prevalence from 3-81%. Only one publication in our study reported the prevalence of fecal incontinence experienced by older Indigenous adults at 2% [57]. A recent systematic review and meta-analysis identified a global prevalence rate of 8.0% [59]. The prevalence of pelvic organ prolapse within the included studies (20-90%) was higher than in other publications. Palmieri et al [14] found a prevalence of 6.1% in a population of pregnant and post-partum Italian women. Other authors report a range from 3.4-10.76% in adult American women [60], 14% in adult Spanish women [15], and 22.7% in Ethiopian women [61]. Rates of dyspareunia identified in our study (24.4-29%) were higher than a sample of women from the United Kingdom at 7.5% [62], but were within the range reported by other authors who explored pregnancy and post-partum (22-44%) [63], and menopause (20-77.6%) [64].

Ethnobotanical and zoological studies were prevalent methodologies in the primary literature and were a high proportion of the interventional publications in this study. These publications described traditional medicine interventions for pelvic health conditions. By mentioning remedies for pelvic health conditions, the studies imply that Indigenous peoples in those groups experience symptoms, although the extent is not clear from the literature.

Emotional experiences such as shame, embarrassment, and stigma were reported by Indigenous peoples. These experiences were in response to having a pelvic health condition and posed a barrier to accessing support and healthcare services. There were numerous barriers to healthcare access identified, which included logistical challenges such as finances and travel requirements. Other reports of healthcare access for Indigenous peoples in Australia [11] and Canada [7,10] include geographic and economic barriers. We also identified pelvic health conditions' impacts on quality of life, relationships, and participation within community. This aligns with existing literature on the negative effects of prolapse [65], pelvic pain conditions [18,66], and bladder incontinence [67] on quality of life.

## Advances

Representation of women is often lacking in biomedical research, and sex and gender is under-reported [68]. The majority of publications in our study reported on sex and gender. There was also a high percentage of women/female participants compared to men/males.

A surprising finding of this review was the number of ethnobotanical and zoological studies. These publications came from the primary literature and reported on the application of traditional knowledge within Indigenous communities to treat pelvic health conditions, by using substances derived from plants and animals. As summarized by Saini [69], Western science has been dismissive of Indigenous ways of knowing, which includes traditional medicines. Additionally, access to Western medicine can be challenging, and Indigenous peoples may turn to traditional remedies, which are more accessible and align with their cultural values [70–73]. Overall, the publications in this review explored the depth of Indigenous knowledges and did not report on the efficacy of the substances.

## Gaps

Among the regions represented in the literature, there were noticeable gaps from regions with known high proportions of Indigenous peoples. The Canadian territories (Northwest Territories, Nunavut, and Yukon) report the highest proportions of Indigenous peoples in the country, at approximately 50% [74], 86% [75], and 22% [76], respectively. Alaska is reported to have the highest proportion of American Indians and Alaska Natives in the United States of America (22%) [77]. However, Alaska and northern Canada were represented with a small number of publications (n = 2). There were also no publications from Iceland, Greenland,

Norway, Sweden, Finland, Russia, several of which are known to be home to Indigenous peoples. Finally, China was missing from the review because the country does not recognize Indigenous populations [78]. Due to this gap, we are unable to make meaningful conclusions about the experiences of Indigenous peoples from these regions, as they may differ from those represented in our dataset.

Upon analysis of the reported age and gender, gaps were also revealed in terms of participant demographics. Although women were represented more in this scoping review, there was a paucity of publications relating to gender diverse Indigenous people. One publication mentioned two-spirit gender identity. In the existing pelvic health literature, there is also a dearth of research with transgender and gender diverse people. However, this field of research is newly expanding, and there are several publications reporting on pelvic health conditions experienced by trans men [79,80] and trans women [81,82]. Additionally, a gap exists in research concerning Indigenous children (0-12 year olds), youth (13-18 year olds), and elders (60+). Pelvic health conditions increase in prevalence as the population ages [15,59], and children and youth can experience different conditions from adults and elders (such as bedwetting and encopresis).

Reporting of certain pelvic health conditions, such as auto-immune diseases (like lichens sclerosis) and bowel conditions was lacking. Perhaps this is due to the low prevalence of autoimmune conditions [83], and known taboos associated with bowel conditions which may inhibit reporting [84,85]. In a systematic review of stigmatized pelvic health conditions by Jouanny et al [86], the authors reported that five of eighty-six articles (6%) studied a bowel condition. This aligns with our finding of 4% of publications reporting bowel conditions.

No publications investigated the economic impacts of pelvic health conditions experienced by Indigenous populations. Other authors have noted a gap in the pelvic health literature with respect to the economic burden of non-cancerous genitourinary conditions in the USA [87]. The authors found that most studies were dated, and few reported aggregate costs. Although outdated, the costs that were reported ranged between $1.2 to $325 billion, depending on the condition [87]. Other publications from the grey literature also report substantial costs to the general population and the systems that serve it. Self-management including medications and products can range from $1400 – 2100 CAD annually for a senior living with incontinence, per The Canadian Continence Foundation [88]. Urinary incontinence alone is estimated to cost the Canadian healthcare system $3.84 billion annually [88]. The Continence Society of Australia estimates that in 2023, incontinence cost Australia $100.4 billion [89].

Finally, few publications mentioned the use of cultural safety and/or trauma-informed approaches. Whether this is due to under-reporting, or under-utilization remains unclear. Limitations in the existing literature may have contributed to these results. Although used widely in healthcare settings, cultural safety within research has not been explored extensively in the existing literature [44]. Gaps also exist in the literature relating to gold standard approaches to trauma-informed research [55].

## Evidence for practice

Pelvic health conditions are prevalent in the general population. As several authors [20–22] have proposed, homogeneity and under-representation of Indigenous people and people of colour in the pelvic health literature may have consequences on clinical decision making. We argue that this may also extend to policy decisions and thereby impact service delivery to Indigenous populations. Although other factors complicate decision making [90–92], the under-representation of Indigenous people in the pelvic health literature further impedes policy development. Decision makers have identified "access and availability of relevant evidence" [90] as a barrier or facilitator to evidence-based decision making. Improving the representation of Indigenous peoples in pelvic health research and sharing those findings with

decision makers will facilitate access to evidence. Additionally, the current literature demonstrates that pelvic health conditions are experienced by Indigenous peoples at comparable prevalence rates to the general population. Recognizing that Indigenous people may live in remote regions [10,11], where delivery of health services can be challenging and costly, decision makers also need to consider novel approaches to increase equity of access.

Discussing pelvic health conditions may be perceived as a private issue, and there may be stigma or cultural norms that health care practitioners should be aware of when working with Indigenous peoples. Creating culturally safe environments within the healthcare system is one way in which practitioners and the systems they work in can improve quality of and access to care. This may be achieved by training in cultural safety; either sought out independently by the practitioner, and/or mandated by the employer or regulatory bodies.

## Recommendations for research

Jefferson et al [55] identified existing gaps with research best practices while working with trauma-exposed populations, and noted the under-representation of marginalized populations. They recommend that gold-standard research guidelines be developed, implemented, and tested, and that trauma-exposed individuals and communities participate in guideline development. Our findings suggest that pelvic health research with Indigenous populations infrequently reports the use of trauma-informed and culturally safe approaches. To better understand how these approaches are used, we recommend that researchers report their use of trauma-informed frameworks and/or culturally safe practices.

Additional research is needed in partnership with Indigenous peoples to co-develop and implement research practices that promote participant safety and are rooted in Indigenous epistemologies. This may be achieved by using community-based, culturally safe, and trauma-informed approaches. Before conducting research with Indigenous communities, we encourage researchers to seek out additional training, frameworks, and resources about research ethics, community engagement, protocols, trauma-informed research, cultural safety, Indigenous methodologies, and community-based research. Some examples include: *Negotiating Research Relationships with Inuit Communities: A Guide for Researchers* [93]*,* University of Manitoba's *Framework for Research Engagement with First Nation, Metis, and Inuit Peoples* [94], or Chapter 9 of the *Tri-Council Policy Statement: Ethical Conduct for Research Involving Humans* [95].

There were disparities in the literature in sex and gender, age, and type of pelvic health conditions reported. Since this review has demonstrated that women and females were more represented, future research in pelvic health conditions should consider greater inclusion of men and gender diverse participants. Additional research is also needed to explore the experiences of Indigenous children, youth, and Elders. Furthermore, under-studied pelvic health conditions included bowel conditions, prolapse, auto-immune disorders, and erectile dysfunction. In regions where there is an abundance of pelvic health research relating to urinary incontinence, for example, we recommend shifting focus towards these under-studied conditions.

## Limitations

Mental wellness and pelvic health conditions are interrelated. Depression and/or anxiety are associated with both urinary and fecal incontinence [67,96,97], interstitial cystitis [66], and pelvic organ prolapse [60]. Women reporting dyspareunia (painful intercourse) can also experience depression, and dissatisfaction or distress with their sex life [62]. We did not actively search for reports or descriptions of mental health diagnoses in our analysis, and as such can make no conclusions about Indigenous peoples' experiences with these often co-occurring conditions.

There are challenges developing a search string representative of Indigenous peoples globally, such as a lack of a universal definition, which leaves researchers to their own devices to develop search terms, resulting in bias and gaps in knowledge [98]. As such, we used the United Nations' description of Indigenous peoples to inform search development and screening criteria. Also, replication of previously used, but inadequate, search terms can perpetuate bias [98]. To address these challenges, and in an attempt to be as comprehensive as possible, a university librarian (SMC) was recruited to develop and run the online database searches. There were other limitations identified during development of the search and sorting of the publications. Firstly, some countries (such as China), do not acknowledge Indigenous peoples' existence within their borders [78] and as such, there is no reporting on their experiences. Secondly, we only included publications in English, which may have resulted in under-reporting from regions that primarily publish in other languages (e.g., Russia, Latin America, etc.). Four full text publications were excluded for this reason, which would account for less than 2% of publications. It is unlikely that our quantitative results would have been impacted significantly. However, we may be missing some context and qualitative experiences unique to these excluded regions. Future reviews may want to consider hiring translators to ensure there is increased regional representation. Finally, some regions (such as Sweden) do not collect data on race or ethnicity [99], which poses barriers to researchers describing and reporting about Indigenous peoples' experiences.

This study demonstrated that Indigenous people experience various pelvic health conditions. It also highlights gaps in the current literature around representation of men and gender diverse people, bowel and auto-immune conditions, and the use of safety frameworks. We provide several recommendations for researchers, clinicians, and decision-makers which may inform future research initiatives, clinical, and health system-level decision making. Several authors [24,26,100] either propose or stress the importance of the final stage of a scoping review: consultation. We agree that this stage is important to contextualize the findings and increase the rigor of the literature review [26]. As discussed in the Recommendations for Research section, research with Indigenous peoples requires additional measures for ethically-sound, culturally safe, and trauma-informed practice. This is a resource-intensive endeavor, and due to the broad scope of this review, the timeline for engagement and consultation falls outside of typical publication timelines. As such, we plan to incorporate the findings of this review into an entire study dedicated to understanding Indigenous women's experiences with pelvic health conditions in northern Canada using community-based participatory principles and a trauma-informed framework. The findings of this work will be shared in future publications.

## Supporting information

**S1 Appendix. Primary literature database search strategies.**
(DOCX)

**S2 Checklist. PRISMA-ScR checklist.**
(DOCX)

**S3 Dataset. Dataset extracted from included full texts.**
(XLSX)

## Author contributions

**Conceptualization:** Kaeleigh Brown, Sandra M. Campbell, Jane Schulz, Pertice Moffitt, Susan Chatwood.

**Data curation:** Kaeleigh Brown, Katherine Choi, Esther Kim.

**Resources:** Sandra M. Campbell.

**Supervision:** Susan Chatwood.

**Visualization:** Kaeleigh Brown.

**Writing – original draft:** Kaeleigh Brown.

**Writing – review & editing:** Kaeleigh Brown, Katherine Choi, Esther Kim, Sandra M. Campbell, Jane Schulz, Pertice Moffitt, Susan Chatwood.

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
