## [Decision Letter · Decision Letter 0]

25 Oct 2024

PONE-D-24-26327Experiences of Indigenous peoples living with pelvic health conditions: A scoping reviewPLOS ONE

Dear Dr. Brown,

Thank you for submitting your manuscript to PLOS ONE. After careful consideration, we feel that it has merit but does not fully meet PLOS ONE’s publication criteria as it currently stands. Therefore, we invite you to submit a revised version of the manuscript that addresses the points raised during the review process.

We look forward to receiving your revised manuscript.

Kind regards,

Mazyar Zahir

Academic Editor

PLOS ONE

Journal Requirements:

"KB received funding support from the AbSPORU Graduate Studentship (https://absporu.ca/) while working on this study. EK was funded by the NWT-Network Environments for Indigenous Health Research (https://cihr-irsc.gc.ca/e/51161.html) as a reviewer. All other authors received no specific funding for this work."

Reviewers' comments:

Reviewer's Responses to Questions

**Comments to the Author**

1. Is the manuscript technically sound, and do the data support the conclusions?

Reviewer #1: Yes

Reviewer #2: Yes

Reviewer #3: Yes

2. Has the statistical analysis been performed appropriately and rigorously? 

Reviewer #1: No

Reviewer #2: Yes

Reviewer #3: Yes

3. Have the authors made all data underlying the findings in their manuscript fully available?

Reviewer #1: Yes

Reviewer #2: Yes

Reviewer #3: Yes

4. Is the manuscript presented in an intelligible fashion and written in standard English?

Reviewer #1: No

Reviewer #2: Yes

Reviewer #3: Yes

5. Review Comments to the Author

Reviewer #1: Experiences of Indigenous peoples living with pelvic health conditions: A scoping review

• The title is clear, concise, and accurately reflects the content of the study.

Abstract

• Include Key Findings: Add a brief mention of the specific patterns or significant results found in the study. This will help readers quickly understand the main contributions of the review.

• Clarify Gaps: Specify the gaps identified in the literature more explicitly in the abstract, such as the underrepresentation of certain populations or regions.

Introduction

• Incorporate Recent References: Include more recent studies or reports (from the last 3-5 years) that highlight the importance of addressing cultural safety and trauma-informed care in Indigenous health research.

• Contextualize Indigenous Health Issues: Expand on the broader context of health disparities faced by Indigenous populations globally, linking these issues directly to the need for focused research on pelvic health conditions.

Methods

• Discuss Protocol Absence: Provide a more detailed discussion on the absence of a registered protocol, including potential impacts on the reproducibility and transparency of the review.

• Address Language Bias: Acknowledge the potential bias introduced by limiting the review to English-language publications. Discuss how this might have affected the inclusivity and comprehensiveness of the review.

• Detail Inclusion/Exclusion Justifications: Provide a more thorough explanation of why certain studies were excluded, especially in the context of Indigenous research, where methodological approaches can be diverse.

Results

• Deeper Analysis of Gaps: Expand on the implications of the identified gaps, particularly the underrepresentation of specific regions (e.g., China, Russia, Nordic countries) and demographics (e.g., gender diverse populations).

• Clarify Exclusion Reasons: In the PRISMA flow diagram and text, clarify the reasons for excluding studies at different stages, especially during full-text screening.

• Enhance Data Presentation: Consider providing more granular data or sub-analyses where possible, such as comparing outcomes across different Indigenous groups or regions.

Discussion

• Link to Policy Implications: Strengthen the connection between the findings and potential policy implications, particularly how they could influence health service delivery and research funding for Indigenous populations.

• Propose Practical Strategies: Offer specific, actionable strategies for incorporating cultural safety and trauma-informed approaches into future pelvic health research, such as recommended training for researchers or guidelines for ethical research practices.

• Expand on Research Gaps: Discuss in more detail why certain conditions (e.g., autoimmune diseases, bowel conditions) are underreported and how future research could address these gaps.

Limitations

• Impact of Limitations: Provide a more in-depth discussion of how the limitations (e.g., language restrictions, lack of data from certain regions) might have influenced the study’s findings and conclusions.

• Consider Alternative Approaches: Suggest alternative strategies that could mitigate the impact of the identified limitations, such as collaboration with researchers fluent in other languages or using broader search terms.

Conclusion

• Strengthen Recommendations: Make the recommendations more forceful, particularly regarding the need for systemic changes in research practices and the importance of addressing identified gaps.

• Next Steps: Briefly outline the next steps for the authors' own research, including any planned studies or initiatives that will build on the findings of this review.

• Highlight Urgency: Emphasize the urgency of addressing the gaps identified, particularly in relation to improving health outcomes for Indigenous populations.

References

• Ensure Currency: Verify that all references are up-to-date, especially those related to recent developments in Indigenous health and pelvic health research.

• Balance of Sources: Ensure a balance between seminal works and recent studies, particularly in areas like cultural safety, trauma-informed care, and Indigenous health disparities.

Overall

• Engage Indigenous Communities: Emphasize the importance of engaging Indigenous communities in the research process, not just as subjects but as active participants in study design, implementation, and dissemination.

• Highlight Ethical Considerations: Discuss the ethical considerations in conducting research with Indigenous populations, including obtaining informed consent, respecting cultural practices, and ensuring that the research benefits the community.

• Consider Visual Enhancements: Add more visual elements, such as tables or infographics, to present complex data more clearly and make the results section more engaging.

With major revisions, particularly in enhancing the depth of analysis, addressing methodological concerns, and strengthening the practical implications, this manuscript has the potential to make a valuable contribution to the field.

Reviewer #2: The manuscript is technically sound, has established methodology and using appropriate descriptive statistics. The data effectively support the conclusions, identifying gaps in the literature and future research needs. The manuscript is well-written in clear, standard English, though some complex data presentation could be simplified for clarity.

I suggest minor revisions as follow:

1- Simplify the presentation of complex prevalence data to enhance accessibility for a broader audience.

2- Consider adding brief explanations or visual aids to clarify the variability in reported rates.

Reviewer #3: Thank you for the opportunity to review this manuscript, and congratulations to the authors for conducting this scoping review. The study extensively reviewed pelvic health conditions in indigenous people worldwide. I found the methodology of the review detailed and acceptable. I just have some comments to clarify a number of issues:

1. The first paragraph of the introduction section should be moved to the end of the introduction, as it states the aim of the study, the lack of this subject in the literature, and its importance.

2. It was stated that “Search terms for pelvic health conditions were informed by the International Continence Society’s Incontinence, 6th Edition…”. We now have the 7th Edition. Therefore, it is suggested to check if the terms were informed by the 7th Edition and update this sentence if it refers to it.

3. I suggest making some minor changes to figure 1 as I have some questions: (A) In line 151, it is written that the results were 2911, but it is 2894 in figure 1. Can the authors explain this discrepancy? I suggest mentioning at the bottom part of figure 1 that 40 publications were finally merged and a total of 203 publications were entered for analysis. (C) It is written that 20 full texts were included via citation searching. However, when we subtract 2032 (excluded abstracts) from 2422 (screened abstracts), we reach 390 full texts assessed for eligibility. What happened to those 20 full texts?

4. Were the Indian subcontinent and the Middle East excluded from Asia in figure 2? Please clarify this.

5. Were bladder control, bladder prolapse, and fistula excluded from the bladder conditions, and was bowel control excluded from the bowel conditions in figure 4? What conditions were labeled as bladder conditions and bowel conditions in this figure?

6. In line 229, it is stated that “Several publications described the influence of the healthcare system and its staff…”. How many were these publications in your review?

6. PLOS authors have the option to publish the peer review history of their article (what does this mean? ). If published, this will include your full peer review and any attached files.

**Do you want your identity to be public for this peer review?** For information about this choice, including consent withdrawal, please see our Privacy Policy .

Reviewer #1: No

Reviewer #2: No

Reviewer #3: No

---

## [Author Response · Author response to Decision Letter 1]

7 Feb 2025

Dear Dr. Zahir and Reviewers,

Thank you for taking the time to review Experiences of Indigenous peoples living with pelvic health conditions: A scoping review, and for providing us with an opportunity to submit a revised manuscript. We appreciate the effort taken by the reviewers to provide in depth and valuable feedback. As outlined below, we have taken the suggested edits and incorporated them into the revised manuscript.

Reviewer 1:

1. Abstract

a. Include Key Findings: Add a brief mention of the specific patterns or significant results found in the study. This will help readers quickly understand the main contributions of the review.

i. See lines 26-33.

b. Clarify Gaps: Specify the gaps identified in the literature more explicitly in the abstract, such as the underrepresentation of certain populations or regions.

i. The patterns and gaps have been described more in the abstract – see lines 26-33.

2. Introduction

a. Incorporate Recent References: Include more recent studies or reports (from the last 3-5 years) that highlight the importance of addressing cultural safety and trauma-informed care in Indigenous health research.

i. We reviewed our citations and updated pelvic health and trauma-informed references to more recent publications.

ii. We expanded more on cultural safety and trauma-informed practice in the methods section. See lines 133-154.

b. Contextualize Indigenous Health Issues: Expand on the broader context of health disparities faced by Indigenous populations globally, linking these issues directly to the need for focused research on pelvic health conditions.

i. An entire summary of health inequities experienced by Indigenous peoples is outside the scope of this manuscript. However, we included some additional wording on what contributes to health inequities, and what is currently known (or not known) about pelvic health conditions experienced by Indigenous peoples. See lines 45-53, and lines 63-75.

3. Methods

a. Discuss Protocol Absence: Provide a more detailed discussion on the absence of a registered protocol, including potential impacts on the reproducibility and transparency of the review.

i. See lines 80-85.

b. Address Language Bias: Acknowledge the potential bias introduced by limiting the review to English-language publications. Discuss how this might have affected the inclusivity and comprehensiveness of the review.

i. This is discussed in the Limitations section (see lines 478-484)

c. Detail Inclusion/Exclusion Justifications: Provide a more thorough explanation of why certain studies were excluded, especially in the context of Indigenous research, where methodological approaches can be diverse.

i. See lines 109-118.

4. Results

a. Deeper Analysis of Gaps: Expand on the implications of the identified gaps, particularly the underrepresentation of specific regions (e.g., China, Russia, Nordic countries) and demographics (e.g., gender diverse populations).

i. We believe the implications of the findings belong in the discussion section, see lines 381-383, 387-390 for more details.

b. Clarify Exclusion Reasons: In the PRISMA flow diagram and text, clarify the reasons for excluding studies at different stages, especially during full-text screening.

i. See lines 168-181, and Table 1.

c. Enhance Data Presentation: Consider providing more granular data or sub-analyses where possible, such as comparing outcomes across different Indigenous groups or regions.

i. We added in additional analysis: publications by year (lines 214-217, Fig 4), regional breakdown of interventional studies (lines 243-255, Fig 5), visualization of prevalence findings (Fig 7), and regional analysis of trauma-informed and culturally safe approaches (Safety heading, lines 233-241).

5. Discussion

a. Link to Policy Implications: Strengthen the connection between the findings and potential policy implications, particularly how they could influence health service delivery and research funding for Indigenous populations.

i. Thank you for pointing this out, especially for researchers that are not familiar with health policy and evidence-based decision making. See lines 418-428.

b. Propose Practical Strategies: Offer specific, actionable strategies for incorporating cultural safety and trauma-informed approaches into future pelvic health research, such as recommended training for researchers or guidelines for ethical research practices.

i. We added in some wording about what topics/training researchers should seek, and some resources to get them started (lines 447-453).

c. Expand on Research Gaps: Discuss in more detail why certain conditions (e.g., autoimmune diseases, bowel conditions) are underreported and how future research could address these gaps.

i. See lines 394-399 for discussion of gaps with respect to certain pelvic health conditions. Future research recommendations are discussed in lines 454-461.

6. Limitations

a. Impact of Limitations: Provide a more in-depth discussion of how the limitations (e.g., language restrictions, lack of data from certain regions) might have influenced the study’s findings and conclusions.

i. See lines 381-383 & 484-485 regarding regional representation, and lines 478-483 regarding language restrictions.

b. Consider Alternative Approaches: Suggest alternative strategies that could mitigate the impact of the identified limitations, such as collaboration with researchers fluent in other languages or using broader search terms.

i. As mentioned for question 3b, with respect to our language limits, see lines 478-483.

ii. Our search terms were developed to be very broad, as evidenced by the number of abstracts (2911) compared to analyzed publications (242). We would be interested to learn what Reviewer 1 had in mind for increasing the breadth of the search.

7. Conclusion

a. Strengthen Recommendations: Make the recommendations more forceful, particularly regarding the need for systemic changes in research practices and the importance of addressing identified gaps.

i. We have provided more actionable recommendations for practice and research.

b. Next Steps: Briefly outline the next steps for the authors' own research, including any planned studies or initiatives that will build on the findings of this review.

i. See lines 496-499.

c. Highlight Urgency: Emphasize the urgency of addressing the gaps identified, particularly in relation to improving health outcomes for Indigenous populations.

i. From our perspective as non-Indigenous practitioners and researchers in women’s health, we agree that addressing these research gaps is important. However, Indigenous communities, governments, and organizations have differing priorities and concerns, and the research community should not be the ones to dictate which research topics are more urgent than others.

8. References

a. Ensure Currency: Verify that all references are up-to-date, especially those related to recent developments in Indigenous health and pelvic health research.

i. We have updated our references to include more recent publications relating to Indigenous health outcomes and trauma-informed practice.

b. Balance of Sources: Ensure a balance between seminal works and recent studies, particularly in areas like cultural safety, trauma-informed care, and Indigenous health disparities.

i. See above note.

9. Overall

a. Engage Indigenous Communities: Emphasize the importance of engaging Indigenous communities in the research process, not just as subjects but as active participants in study design, implementation, and dissemination.

i. Engagement and collaboration with Indigenous peoples is very important. We believe we have touched on the importance of this in lines 444-453.

b. Highlight Ethical Considerations: Discuss the ethical considerations in conducting research with Indigenous populations, including obtaining informed consent, respecting cultural practices, and ensuring that the research benefits the community.

i. We did not further elaborate on the ethical considerations of conducting research with Indigenous peoples, as we believe it falls outside the scope of this manuscript. Instead, we directed readers to several resources. See lines 450-453.

c. Consider Visual Enhancements: Add more visual elements, such as tables or infographics, to present complex data more clearly and make the results section more engaging.

i. Added three additional figures (Fig 4, Fig 5, Fig 7).

Reviewer #2:

1. Simplify the presentation of complex prevalence data to enhance accessibility for a broader audience.

a. See Fig 7 for floating bar graph.

2. Consider adding brief explanations or visual aids to clarify the variability in reported rates.

a. See Fig 7 for floating bar graph. Lines 272-273, and 277-278 provide a high-level overview of the prevalence findings, leaving the more detailed representation to the figure.

Reviewer #3:

1. The first paragraph of the introduction section should be moved to the end of the introduction, as it states the aim of the study, the lack of this subject in the literature, and its importance.

a. See lines 68-75 for the recommended changes.

2. It was stated that “Search terms for pelvic health conditions were informed by the International Continence Society’s Incontinence, 6th Edition…”. We now have the 7th Edition. Therefore, it is suggested to check if the terms were informed by the 7th Edition and update this sentence if it refers to it.

a. Search terms were developed in 2022, when the 6th edition was the most recent version. We clarified this in the manuscript (see lines 94-96). We did not use the 7th edition.

3. I suggest making some minor changes to figure 1 as I have some questions: (A) In line 151, it is written that the results were 2911, but it is 2894 in figure 1. Can the authors explain this discrepancy? I suggest mentioning at the bottom part of figure 1 that 40 publications were finally merged and a total of 203 publications were entered for analysis. (C) It is written that 20 full texts were included via citation searching. However, when we subtract 2032 (excluded abstracts) from 2422 (screened abstracts), we reach 390 full texts assessed for eligibility. What happened to those 20 full texts?

a. Thanks for catching this error! Covidence’s PRISMA diagram was used to develop Fig 1, and the program sorts duplicates differently than how we presented it. Fig 1 has now been updated to be accurate. We also added more description in the text (see lines 168-181).

4. Were the Indian subcontinent and the Middle East excluded from Asia in figure 2? Please clarify this.

a. Yes, see lines 186-188 for an explanation.

5. Were bladder control, bladder prolapse, and fistula excluded from the bladder conditions, and was bowel control excluded from the bowel conditions in figure 4? What conditions were labeled as bladder conditions and bowel conditions in this figure?

a. See lines 261-269 for more detail of what was included in each category.

6. In line 229, it is stated that “Several publications described the influence of the healthcare system and its staff…”. How many were these publications in your review?

a. Number of publications updated, see 287.

Formatting & Funding Disclosures

a. Added indent to beginning of each paragraph.

b. Modified Table 1 to align with formatting requirements.

c. Removed author contributions and ORCID from title page.

d. Removed bold formatting on supplemental material, figures, and table citations.

e. Confirmed that font in the figures is Arial, size 8-12pt.

f. Altered colours in figures to grey scale.

g. Added legends to applicable figures (Fig 2-4).

h. Reviewed reference list to comply with Vancouver style citations

"KB received funding support from the AbSPORU Graduate Studentship (https://absporu.ca/) while working on this study. EK was funded by the NWT-Network Environments for Indigenous Health Research (https://cihr-irsc.gc.ca/e/51161.html) as a reviewer. All other authors received no specific funding for this work." The funders had no role in study design, data collection and analysis, decision to publish, or preparation of the manuscript.

a. A revised funding statement is included at the end of our cover letter. I have confirmed with all co-authors about their funding, and have updated the statement.

Again, thank you for providing us with suggestions on how to improve this manuscript, and taking time out of your busy schedules to do so.

We look forward to hearing from you regarding this submission and to respond to any further questions and comments you may have.

Sincerely,

Kaeleigh Brown

---

## [Decision Letter · Decision Letter 1]

26 Feb 2025

Experiences of Indigenous peoples living with pelvic health conditions: A scoping review

PONE-D-24-26327R1

Dear Dr. Brown,

We’re pleased to inform you that your manuscript has been judged scientifically suitable for publication and will be formally accepted for publication once it meets all outstanding technical requirements.

Kind regards,

Mazyar Zahir, MD

Academic Editor

PLOS ONE

Additional Editor Comments (optional):

<small><var>Thank you very much for submitting your valuable research to our journal. We look forward to your future submissions. </var></small>

<small><var>The only revision that I would like to see during proofreading and before publication pertains to your 88th reference: </var></small><small><var>"The Canadian Continence Foundation. The impact of incontinence in Canada: A briefing document for policy-makers [Internet]. 2014 Dec [cited 2024 Jun 26]. Available from: https://www.canadiancontinence.ca/pdfs/en-impact-of-incontinence-in-canada-2014.pdf". </var></small><small><var>As far as I am aware, the link provided in this reference does not lead to a valid document and should be reviewed before publication</var></small>.

Reviewers' comments:

Reviewer's Responses to Questions

**Comments to the Author**

1. If the authors have adequately addressed your comments raised in a previous round of review and you feel that this manuscript is now acceptable for publication, you may indicate that here to bypass the “Comments to the Author” section, enter your conflict of interest statement in the “Confidential to Editor” section, and submit your "Accept" recommendation.

Reviewer #1: All comments have been addressed

Reviewer #2: All comments have been addressed

Reviewer #3: All comments have been addressed

2. Is the manuscript technically sound, and do the data support the conclusions?

Reviewer #1: Yes

Reviewer #2: Yes

Reviewer #3: (No Response)

3. Has the statistical analysis been performed appropriately and rigorously? 

Reviewer #1: Yes

Reviewer #2: Yes

Reviewer #3: (No Response)

4. Have the authors made all data underlying the findings in their manuscript fully available?

Reviewer #1: (No Response)

Reviewer #2: Yes

Reviewer #3: (No Response)

5. Is the manuscript presented in an intelligible fashion and written in standard English?

Reviewer #1: Yes

Reviewer #2: Yes

Reviewer #3: (No Response)

6. Review Comments to the Author

Reviewer #1: (No Response)

Reviewer #2: (No Response)

Reviewer #3: (No Response)

7. PLOS authors have the option to publish the peer review history of their article (what does this mean? ). If published, this will include your full peer review and any attached files.

**Do you want your identity to be public for this peer review?** For information about this choice, including consent withdrawal, please see our Privacy Policy .

Reviewer #1: No

Reviewer #2: No

Reviewer #3: No

---

## [Editor Report · Acceptance letter]

PONE-D-24-26327R1

PLOS ONE

Dear Dr. Brown,

I'm pleased to inform you that your manuscript has been deemed suitable for publication in PLOS ONE. Congratulations! Your manuscript is now being handed over to our production team.

Kind regards,

on behalf of

Dr. Mazyar Zahir

Academic Editor

PLOS ONE